# Emotional Availability in Autism Intervention: A Mother–Father Comparative Analysis

**DOI:** 10.3390/brainsci15020133

**Published:** 2025-01-29

**Authors:** Silvia Perzolli, Giulio Bertamini, Paola Venuti, Arianna Bentenuto

**Affiliations:** 1Laboratory of Observation, Diagnosis, and Education (ODFLab), Department of Psychology and Cognitive Science, University of Trento, 38068 Rovereto, Italy; silvia.perzolli@unitn.it (S.P.); giulio.bertamini@unitn.it (G.B.); arianna.bentenuto@unitn.it (A.B.); 2Department of Child and Adolescent Psychiatry, Pitie-Salpetriere University Hospital, Sorbonne University, 75013 Paris, France

**Keywords:** autism, parents, parental intervention, emotional availability

## Abstract

Background/Objectives: The literature highlights the importance of parental involvement in autism treatment. However, much research has predominantly focused on child outcomes and cognitive dimensions. This study explores the impact of an early intensive intervention with parental involvement, focusing on changes in parents’ affective exchanges. Notably, given the paucity of studies on fathers in the intervention context, this study examines the comparative trajectory of change considering both caregivers. Methods: Twenty autistic preschoolers were monitored for one year during a parental-based intervention. Child–mother and child–father play interactions were coded with the Emotional Availability Scales at baseline and at 12 months. Repeated measures linear mixed-effect models were employed to investigate time and caregiver effects and their interaction. Results: Results highlighted both similarities and differences in change trajectories between caregivers. Parental sensitivity, structuring, and non-intrusiveness significantly increased for both parents with fathers showing more prominent gains in structuring the interaction while being non-intrusive. Child responsiveness and involvement significantly increased, showing similar trajectories with both caregivers. Children were generally more involved while interacting with their fathers. Conclusion: Parent–child interactions with caregivers evolved toward more adaptive exchanges regarding emotional availability for children’s and parents’ dimensions. Fathers appeared to be particularly receptive regarding acquiring structuring abilities and non-intrusive behaviors. Our results underscore the importance of investigating parental features as well as the importance of actively involving caregivers to support distal outcomes and generalization.

## 1. Introduction

Parents are biologically inclined to intuitively interpret and respond to children’s typical developmental signals [1]. However, for autistic children, parents often face challenges in decoding their signals and addressing their unique needs effectively, potentially hindering successful parenting approaches. Given the critical influence of parental qualities on child developmental outcomes, recent findings strengthen the importance of involving caregivers during intervention to increase dyadic syntonization levels and extend the acquisition of competencies in naturalistic contexts (e.g., home) [2,3,4,5]. Currently, many international and national guidelines [6,7] recommend parental inclusion during intervention with autistic children. A substantial body of research showed that parental involvement during the intervention improves child outcomes in areas such as communication, social development [8], and symptom reduction [9] while fostering the generalization of skills across settings [10]. Some research showed that without involving caregivers, child’s outcome variables tend to remain more stable over time [8,11]. Additionally, research suggests that engaging parents helps overcome potential barriers created by parental difficulties in social communication, leading to improved caregiving strategies. Interestingly, studies have shown that improvements in parent–child interaction correlate with better developmental outcomes for children [12], further emphasizing the importance of dyadic relational processes. Despite this evidence, much of the research in this field focuses predominantly on child outcomes, which are often assessed through standardized tools such as Autism Diagnostic Observation Schedule (ADOS-2) and Griffiths Scales (GMDS-ER) [13,14,15,16,17,18,19]. While these measures are valuable, they may lack sensitivity to detect subtle changes in social communication and interaction occurring during interventions. Observational and behavioral tools, such as the Emotional Availability Scales (EA Scales, [20]), offer a more dynamic approach to understanding dyadic interaction by capturing relational nuances between parents and children [21]. In line with this, analyzing the dyad’s interactive component through observational and behavioral measures might provide a sensitive-to-change perspective to investigate caregiver and child improvements in the relational context.

### 1.1. Mothers’ Involvement in Intervention

Research on parental involvement in interventions has traditionally focused more on mothers than fathers. Mothers of autistic children often demonstrate diverse responses during interventions. Some studies suggest that they exhibit similar sensitivity levels as mothers of typically developing children [22], while others report lower sensitivity and difficulties in timing and quality of interactions [23]. The observative instruments used in previous studies were proven to be particularly sensitive in capturing mothers’ characteristics, but they did not adequately capture children’s responses to their mothers’ interactive exchanges. The Emotional Availability (EA) Framework evaluates the quality of emotional exchanges in parent–child dyads and is particularly relevant to understanding maternal contributions during interventions and children’s responses. EA captures critical aspects such as warmth, sensitivity, and mutual responsiveness, making it a powerful tool for assessing changes in mother–child dynamics over time [24]. Meanwhile, research has demonstrated the predictive value of EA for positive developmental outcomes, such as attachment security [25] and emotional regulation [26]. The construct of Emotional Availability (EA) provides a scientifically grounded framework for understanding the quality of connection and communication between caregivers and children. It captures the dyadic capacity to form a mutually fulfilling, healthy relationship and to share an emotional connection [24]. Rooted in the integration of attachment theory [27], the theory of emotions [28], and maternal sensitivity [29], EA offers a multidimensional approach to studying caregiver–child interactions. In their seminal work, [30] emphasized the multidimensional nature of interaction styles, the regulatory role of emotions within these exchanges, and the dyadic perspective that considers contributions from both partners. This framework addresses positive and negative relational patterns and demonstrates applicability across diverse contexts [31]. To our knowledge, no longitudinal studies investigated EA in the context of intervention with both mothers and fathers. Addressing this gap is essential to tailoring interventions in autism that enhance maternal and paternal sensitivity and responsiveness, thereby improving outcomes for parents and children.

### 1.2. Fathers’ Involvement in Intervention

Despite robust evidence highlighting fathers’ contributions to child development and family well-being, their role in intervention remains unexplored [32,33,34]. Parenting interventions have historically prioritized maternal involvement, often neglecting fathers [35]. Furthermore, evaluations of father participation are stymied by how parental interventions are currently designed, delivered, and evaluated, explicitly focusing on maternal characteristics and behaviors compared to paternal ones. Recent studies underscore the unique benefits of engaging fathers during intervention. For instance, paternal involvement has been associated with improved child outcomes, including enhanced play skills, language development, and social engagement [32]. Fathers also report more effective interactions with their children following targeted training [36]. However, much of the existing research relies on self-reported measures or qualitative interviews with no studies investigating the father–child affective quality during intervention through highly standardized dyadic and observative tools [32]. Including EA Scales in studying father–child dyads could offer valuable insights into parental contributions during interventions. By focusing on affective quality and mutual responsiveness, EA can help identify specific strategies to enhance fathers’ interactions with their children, promoting better outcomes and understanding paternal-specific characteristics. In addition, by focusing mainly on the maternal role, researchers and practitioners may unintentionally burden mothers with greater responsibilities that lead to higher maternal stress and a negative impact on the child [37,38]. Expanding research on fathers in intervention contexts is crucial for fostering a more balanced approach to parenting and for optimizing the developmental benefits for autistic children.

### 1.3. The Current Study

In accordance with the above, the present work aimed to assess changes in an intensive intervention with parent involvement, focusing on caregivers and dyadic outcomes, through one standardized observational instrument that allows the evaluation of affective quality (Emotional Availability Scales, EAS [20]).

First, in the present study, we intended to investigate the longitudinal changes of specific interactive modalities during the intervention for mothers and fathers. Second, we aimed to explore how a parental-based intervention impacts the interactive characteristics of the dyad and how this relationship evolves differently for mothers and fathers over time.

On this basis, we hypothesized as follows:In line with previous findings that include caregivers in the therapeutic setting [9,10] and with the theoretical framework of the implemented intervention, which focuses on the syntonization between adult and child needs, we expected that both mothers and fathers would increase their interactive abilities. Given the possibility for parents to experiment with functional interaction with their children during intervention, we particularly expected an increase in parental awareness of timing, the ability to catch child signals, and respect for their timing.In line with previous research about the domain of non-hostility [39] and prior studies assessing changes during the intervention [40], we did not expect significant variations in this dimension given its stability over time.Furthermore, considering that therapists provide caregivers with appropriate guidance and suggestions to interact and play with their children during the intervention, we also predicted that both parents will increase their general levels of EA. We hypothesized that the structuring and non-intrusiveness scales would particularly highlight the most prominent changes given their closer relationship with the purposes of parental-based intervention. In addition, given that mothers often display different interactive styles compared to fathers, as pointed out by the literature on typical development [41,42], we hypothesized different patterns of changes.For the child, in line with previous findings that depicted a positive change in child socio-communicative behaviors [43,44], we expected to find improvements in the child’s level of responsiveness and the use of different communicative strategies (e.g., eye-contact looking, body positioning, verbal involvement) to involve both caregivers and participate in the interactive exchange.

## 2. Materials and Methods

### 2.1. Participants and Procedure

This study involved 40 parents, consisting of 20 mothers (mean age = 36.45 years, SD = 2.96) and 20 fathers (mean age = 38.25 years, SD = 4.011) of 20 autistic preschool children (18 males and 2 females) (mean chronological age = 38.25 months, SD = 11.84; mean mental age = 29.74 months, SD = 13.02). All data were collected at ODFLab (Laboratory of Observation, Diagnosis, and Education) of the Department of Psychology and Cognitive Science, University of Trento (Rovereto Italy). ODFLab is a research and clinical center where families or individuals can receive functional diagnoses for themselves or their children and/or participate in therapeutic programs. The investigation complied with the last version of the Declaration of Helsinki [45]. The Ethics Committee of the University of Trento (Italy) approved all procedures (protocol number 2020-042). Both parents participated in an early intensive treatment program that included parent involvement (see paragraph below for details). The diagnosis of ASD was confirmed through clinical judgment by an independent clinician based on the DSM-5 criteria for Autism Spectrum Disorder as well as through the administration of the Autism Diagnostic Observation Schedule (ADOS-2; [46]). Furthermore, the Griffiths Mental Development Scale Edition Revised (GMDS-ER; [47]) assessed children’s general developmental quotient and mental age.

Participants were recruited voluntarily through advertisements displayed in the laboratory waiting room. A dedicated meeting was scheduled with a researcher not involved in the clinical process to explain the research objectives and procedure. Finally, families signed a written consent form to participate in the study.

The Emotional Availability Scales (EAS, [20]) were used to assess the affective quality within the dyad and applied to ten-minute video recordings of mother–child and father–child interactions acquired during the first diagnostic assessment before intervention. A set of toys, selected according to each child’s chronological age, was used for these interactions, including a train, toy car, toy phone, dinette set made of glasses, cutlery, mocha, mugs, saucers and pans, doll, ball, puzzle box, and books. During these recordings, both parents were instructed to play spontaneously and autonomously with their children as they would in a naturalistic setting. After one year of intervention, a second assessment of the child’s functioning was conducted, and new video-recorded interactions between mother–child and father–child dyads were collected. We considered a one-year interval, since the clinical practice and experience indicates that a 12-month period provides a reasonable timeframe to capture subtle changes in this domain, considering the structural framework and design of the intervention described in the Methods section.

Two independent observers trained in applying the instrument were randomly assigned to code the EAS. The observers reached a significant level of interrater reliability. A two-way mixed effects intraclass correlation coefficient (ICC) with absolute agreement was used [48]. Additionally, the two observers coded four target videos (total duration 120 min) as part of their training. The interrater reliability coefficients ranged from 0.84 to 0.92.

### 2.2. Measures

#### 2.2.1. Emotional Availability Scales

The Emotional Availability Scales (EAS; [20]) were developed to operationalize the EA construct through observational measures. The scales include four dimensions for adults (sensitivity, structuring, non-hostility, and non-intrusiveness) and two for children (responsiveness and involvement). Scoring is conducted on a 1 to 7 Likert scale with additional subscales scored from 1 to 3 for more specific dimensions. Midpoint scoring, particularly for children with disabilities, is strongly recommended and supported by previous research [39]. Guidelines for applying the EAS to children with Developmental Disabilities (DDs) and Autism Spectrum Disorder (ASD) have also been established. Adult Sensitivity captures the caregiver’s ability to perceive and respond appropriately to the child’s communicative signals while maintaining emotional attunement. Structuring refers to the caregiver’s capacity to scaffold the child’s activities with appropriate prompts, fostering autonomy. Non-intrusiveness evaluates the caregiver’s ability to support interactions without being overbearing, while Non-Hostility measures the absence of overt (e.g., aggression) or covert (e.g., boredom, detachment) hostility. On the child side, Responsiveness assesses the frequency and quality of the child’s positive responses to caregiver bids. At the same time, Involvement evaluates the child’s efforts to engage the caregiver through modalities such as eye contact, verbal expressions, and body language. This multidimensional framework underscores the critical role of emotional dynamics in caregiver–child interactions and provides a robust tool for advancing research and intervention in diverse developmental contexts. Optimal levels of Emotional Availability (EA; score of 7) are defined by high-quality affective exchanges, robust parental support, and the child’s exceptional responsiveness and involvement. Moderate levels (scores of 5–6) reflect effective parental interaction patterns, where the caregiver demonstrates appropriate but not ideal strategies for engaging with the child. Apparent or inconsistent levels of EA (score of 4) indicate inconsistency in the caregiver’s ability to guide the child accompanied by the child displaying a mix of positive and negative strategies to gain the caregiver’s attention. Low (scores of 3/2.5) or very low levels (scores of 2–1) of EA are associated with caregivers who exhibit insensitivity and emotional unavailability. In these cases, children often display signs of worry, anxiety, and distress with their strategies for responsiveness and involvement either being inappropriate or absent.

#### 2.2.2. Autism Diagnostic Observation Schedule-2 (ADOS-2)

The severity of a child’s symptoms was evaluated before and after the intervention using the Autism Diagnostic Observation Schedule, Second Edition (ADOS-2; [46]), which is a gold-standard instrument for ASD diagnosis. Administration of the ADOS-2 is conducted by trained psychologists who have completed an official certification course. The tool offers different modules tailored to the child’s chronological age and expressive language level. Each module generates a total score for diagnostic classification into one of three categories: Autism, Autism Spectrum, or Non-Spectrum. This total score is then converted into a comparison score, allowing cross-module comparisons and categorizing symptom severity into mild, moderate, or severe levels.

#### 2.2.3. Griffiths Mental Development Scales-III Edition Revised (GMDS-ER)

Child cognitive development before and after the intervention was evaluated using the Griffiths Mental Development Scales—Extended Revised (GMDS-ER; [47]). The GMDS-ER comprises standardized developmental scales, including norms for an Italian population. They are administered in a laboratory setting through semi-structured activities designed to assess various dimensions of mental development in infants and children. These scales generate Z-scores across six developmental domains: Locomotion, Personal–Social, Communication and Listening, Eye–Hand Coordination, Performance, and Practical Reasoning. The GMDS-ER provides a General Quotient (GQ) and a developmental age-equivalent to identify potential developmental delays and specific quotients and age-equivalents for each subscale. All scores are standardized with a mean of 100 and a standard deviation of 15.

#### 2.2.4. Parental-Based Intervention

The “Italian Model of Intervention” at ODFLab integrates developmental, relationship-based, and behavioral principles within the framework of the Italian Health System [7,49,50,51], which was inspired by the Early Start Denver Model [43,52] and the Preschool Autism Communication Trial [8]. This approach emphasizes parent–child reciprocity, intersubjective exchanges, and the child’s intentionality. The intervention includes caregiver involvement to promote functional interactions, improve emotional availability, and support the child’s skill generalization in naturalistic contexts. Unlike parent-mediated programs, parents are not required to deliver the intervention but instead engage in sessions structured by therapists, allowing for increased dyadic synchrony, parental self-efficacy, and reduced stress. Tailored interventions address three levels: relationship (e.g., synchrony in parent–child interactions), behavior (e.g., augmentative communication), and development (e.g., music therapy, cognitive activation, occupational therapy), which were adjusted to the child’s profile. Structured phases transition from intensive intervention (6–8 h per week with individual and parent–child sessions) to consolidation (group interventions for social play and biweekly parent support), which were followed by ongoing availability during critical transitions or challenges. Additionally, parental interventions focus on reflective practices, interactive video feedback, and peer support, empowering caregivers to address maladaptive interaction patterns while fostering sensitivity and responsiveness to their child’s needs. Supervision is provided monthly by licensed psychotherapists to ensure fidelity and efficacy.

### 2.3. Statistical Analyses

First, descriptive statistics were calculated (See Table 1). Then, given the repeated measure design with the caregiver as a grouping variable, Linear Mixed Models (LMMs, package lme4) were used to address the research question. Specifically, the fixed effects of time and caregiver were considered together with their interaction. Fixed effects allowed the evaluation of significant changes over time in independent variables of interest and whether interacting with mothers showed significant differences with respect to fathers. More importantly, the interaction term allowed us to test whether changes over time differed between the two caregivers as well as identify the presence of different profiles and modalities of change during intervention. A random effect was also included to account for variability across children, modeling an intercept for each child. LMMs are also suitable for non-normally distributed data. Models were checked for assumptions of linearity, normality of residuals and random effects (Shapiro–Wilk test), and homoscedasticity (Levene test). Cluster-robust variance–covariance estimates with bias-reduced linearization adjustment [53] of model parameters were reported to mitigate the reduced sample size and provide robust estimates. Models were fitted using restricted maximum likelihood (REML) and evaluated by marginal and conditional R2. Multicollinearity was addressed by the Variance Inflation Factor (VIF). Paired Cohen’s *d* were also reported for the main effects of time and caregiver accounting for within-subjects correlation.

Additionally, models were compared with a simpler version without the caregiver–time interaction through an ANOVA with a Chi-squared test. Aiken Information Criteria (AIC) was also used for model comparison. Finally, model coefficients were bootstrapped (*N* = 10,000) to produce Confidence Intervals (CIs) and to test for model robustness. Statistical analyses were conducted with R 3.6.3.

## 3. Results

### Linear Mixed Models

Concerning caregiver EAS dimensions, the first model was fitted with Sensitivity as a dependent variable. The intercept was significant (b = 4.90; SE = 0.19; *t* (19) = 25.46; *p* < 0.001; CI = [4.58,5.17]). There was no fixed effect of caregiver (b = 0.03; SE = 0.19; *t* (19) = 0.13; *p* = 0.90; CI = [−0.33,0.38]). Conversely, the fixed effect of time was significant (b = 0.43; SE = 0.13; *t* (19) = 3.34; *p* = 0.003; CI = [0.08,0.78]). The caregiver–time interaction did not reach statistical significance (b = 0.33; SE = 0.19; t (19) = 1.75; *p* = 0.10; CI = [−0.17,0.81]). The VIF for all model terms was lower than 3, indicating no multicollinearity issues. Therefore, mothers and fathers were equally emotionally attuned while interacting with their children. Further, caregiver sensitivity significantly increased over time similarly between fathers and mothers. Marginal and conditional R2 values were 0.18 and 0.44, respectively, indicating that fixed effects alone explained 18% of the observed variance. Cohen’s d values for caregiver and time were 0.23 [−0.08, 0.55] and 0.82 [0.46, 1.18], respectively. The ANOVA did not reach statistical significance (Chisq(1) = 1.71; *p* = 0.19), indicating that the simpler model (AIC = 172) without the interaction term better fit the data compared to the full model (AIC = 173).

The second dimension was EAS parent structuring abilities. Model intercept was significant (b = 4.63; SE = 0.21; *t* (19) = 22.24; *p* < 0.001; CI = [4.31,4.94]). There was no fixed effect of caregiver (b = −0.15; SE = 0.19; *t* (19) = −0.81; *p* = 0.43; CI = [−0.49,0.19]). Conversely, the fixed effect of time was significant (b = 0.33; SE = 0.12; *t* (19) = 2.67; *p* = 0.02; CI = [0,0.66]). The caregiver–time interaction was also significant (b = 0.70; SE = 0.21; *t* (19) = 3.39; *p* = 0.003; CI = [0.23,1.16]). The VIF for all model terms was lower than 3, indicating no multicollinearity issues. Hence, mothers and fathers showed comparable structuring abilities while interacting with their children. Moreover, their structuring capacities significantly increased over time during intervention for both parents. Fathers also showed significantly higher improvements compared to mothers. Marginal and conditional R2 values were 0.23 and 0.58, respectively, indicating that fixed effects alone explained 23% of the observed variance. Cohen’s d for caregiver and time were 0.23 [−0.08, 0.55] and 0.90 [0.53, 1.27], respectively. The ANOVA was also significant (Chisq(1) = 8.36; *p* = 0.004), indicating that the model with the interaction term (AIC = 173) better fit the data with respect to the simpler one (AIC = 178), further underscoring the presence of distinct trajectories of change between the two caregivers.

The third caregiver model included EAS Non-Intrusiveness. The model intercept was significant (b = 4.83; SE = 0.21; *t* (19) = 23.41; *p* < 0.001; CI = [4.50,5.14]). There was no fixed effect of caregiver (b = −0.08; SE = 0.19; *t* (19) = −0.40; *p* = 0.70; CI = [−0.42,0.27]). Conversely, the fixed effect of time was significant (b = 0.30; SE = 0.11; *t* (19) = 2.70; *p* = 0.01; CI = [−0.03,0.64]). The caregiver–time interaction was also significant (b = 0.50; SE = 0.19; *t* (19) = 2.70; *p* = 0.01; CI = [0.02,0.97]). The VIF for all model terms was lower than 3, indicating no multicollinearity issues. Mothers and fathers showed comparable non-intrusive behaviors in interaction with their children. Moreover, this ability significantly increased over time during intervention for both caregivers. In addition, fathers showed significantly higher improvements with respect to mothers. Marginal and conditional R2 were 0.15 and 0.54, respectively, indicating that fixed effects alone explained the 15% of the observed variance. Cohen’s d for caregiver and time were 0.21 [−0.11,0.52] and 0.82 [0.46,1.18], respectively. The ANOVA was significant (Chisq(1) = 4.24; *p* = 0.04), indicating that the model with the interaction term (AIC = 176) better fit the data compared to the simpler model (AIC = 177).

Finally, caregiver EAS Non-Hostility was considered as a dependent variable. The model intercept was significant (b = 5.65; SE = 0.15; *t* (19) = 38.80; *p* < 0.001; CI = [5.38,5.92]). There was no fixed effect of caregiver (b = −0.07; SE = 0.11; *t* (19) = −0.68; *p* = 0.51; CI = [−0.31,0.16]). As well, the fixed effect of time was not significant (b = 0.10; SE = 0.12; *t* (19) = 0.85; *p* = 0.41; CI = [−0.13,0.34]). The caregiver–time interaction was also non-significant (b = 0; SE = 0.15; *t* (19) = 0; *p* = 1; CI = [−0.33,0.32]). The VIF for all model terms was lower than 3, indicating no multicollinearity issues. Father and mothers showed comparable non-hostility levels during the interaction. This ability did not significantly increase over time, and trajectories were similar for both parents. Marginal and conditional R2 were 0.01 and 0.62, respectively, indicating that fixed effects alone did not explain the observed variance. Cohen’s d values for caregiver and time were 0.14 [−0.45,0.18] and 0.21 [−0.11,0.52], respectively. The ANOVA was non-significant (Chisq(1) = 0; *p* = 1), indicating that the simpler model without the interaction term (AIC = 127) better fit the data compared to the full model (AIC = 130).

For child EAS dimensions, the first model included Responsiveness as a dependent variable. The intercept was significant (b = 3.60; SE = 0.18; *t* (19) = 20.01; *p* < 0.001; CI = [3.29,3.91]). The fixed effect of caregiver was not significant (b = 0.13; SE = 0.15; *t* (19) = 0.82; *p* = 0.43; CI = [−0.13,0.38]). Conversely, there was a significant fixed effect of time (b = 0.58; SE = 0.11; t (19) = 5.21; *p* < 0.001; CI = [0.33,0.83]). The caregiver–time interaction was not significant (b = 0.10; SE = 0.17; *t* (19) = 0.58; *p* = 0.57; CI = [−0.26,0.45]). The VIF for all model terms was lower than 3, indicating no multicollinearity issues. Therefore, no significant differences emerged concerning the caregiver role, and child responsivity significantly increased over time, similarly with both parents. Marginal and conditional R2 were 0.17 and 0.73, respectively, indicating that fixed effects alone explained 17% of the observed variance. Cohen’s d for caregiver and time were 0.30 [−0.02,0.62] and 1.16 [0.75,1.55], respectively. The ANOVA was not significant (Chisq(1) = 0.32; *p* = 0.57), indicating that the simpler model (AIC = 142) without the interaction term better fit the data compared to the full model (AIC = 145).

The last model tested EAS Child Involvement as a dependent variable. The intercept was significant (b = 3.10; SE = 0.16; *t* (19) = 19.30; *p* < 0.001; CI = [2.83,3.37]). The fixed effect of caregiver was significant (b = 0.23; SE = 0.08; *t* (19) = 2.65; *p* = 0.02; CI = [0,0.46]). There was also a significant fixed effect of time (b = 0.40; SE = 0.09; *t* (19) = 4.29; *p* < 0.001; CI = [0.17,0.63]). The caregiver–time interaction did not reach statistical significance (b = 0.30; SE = 0.17; *t* (19) = 1.75; *p* = 0.10; CI = [−0.03,0.62]). The VIF for all model terms was lower than 3, indicating no multicollinearity issues. Therefore, a significant difference emerged for caregivers: children were generally more involved while interacting with their fathers with respect to their mothers. Furthermore, child involvement significantly increased over time similarly with both parents. Marginal and conditional R2 values were 0.23 and 0.73, respectively, indicating that fixed effects alone explained 23% of the observed variance. Cohen’s d values for caregiver and time were 0.71 [0.36,1.05] and 0.98 [0.59,1.35], respectively. The ANOVA did not reach statistical significance (Chisq(1) = 3.31; *p* = 0.07), indicating that the simpler model (AIC = 129) without the interaction term better fit the data compared to the full model (AIC = 130).

The EAS model interaction plots are shown in Figure 1.

## 4. Discussion

In recent years, several studies have examined behavioral changes in children’s outcomes within parental-based interventions [9,12]. However, less attention has been paid to children’s emotional changes and even less to dyadic changes during intervention particularly considering both mothers and fathers. Exploring interactive longitudinal profiles may improve our understanding of specific aspects of mothers and fathers to orient and tailor interventions, leveraging their strengths and supporting their areas for improvement.

This study aimed to analyze longitudinal changes in specific dimensions of the exchange after intervention with parents. To accomplish that, we relied on the construct of Emotional Availability to measure the dimensions of interaction quality. We employed a repeated measure design that could investigate longitudinal trajectories by considering both caregivers. Our findings highlighted a general improvement in the dyadic interactive characteristics. We also found similarities and differences in the trajectories of mothers and fathers.

With respect to the scale of Parental Sensitivity, results indicated that mothers and fathers showed both similar levels and trajectories of change in terms of sensitivity. Before the intervention, they showed low–medium scores, whereas after intervention, they received medium scores, indicating better attunement to the child’s needs during play for both parents. In particular, both parents seemed to display more appropriate and prompt responses to their child’s signals, demonstrating greater awareness of timing to support interactions.

Considering the Non-Hostility scale, we found no significant changes before or after intervention for either parent. In addition, parents obtained scores in the good range, indicating the absence of aggressive and hostile verbal and nonverbal behaviors directed at their children. Other studies [39] have also reported the stability of this scale, indicating that some of its elements (e.g., threats of separation, frightening behaviors, etc.) are unlikely to occur during naturalistic playful exchanges.

Regarding the scale of Structuring, our results indicated that both parents improved significantly over time in their abilities to provide guidance and suggestions to their children during the interaction. Interestingly, we also found a significant difference in the longitudinal changes of parents. Although both had low–medium scores at the beginning of intervention, mothers remained in the same range, whereas fathers obtained scores in the medium–good range after the intervention. A similar pattern has been found for the Non-Intrusiveness scale, indicating that both parents showed increased abilities in following the child’s lead. Notably, also in this case, fathers showed more prominent gains compared to mothers.

Taken together, these results suggest that fathers and mothers display both similarities and differences in their interactive profiles. Both caregivers are characterized by significant and similar improvements in the affective dimension of Sensitivity. However, cognitive dimensions like Structuring and Non-Intrusiveness ability show different trajectories of change between parents with fathers achieving more prominent gains during intervention. This preliminary result seems to align with the literature that underscores a paternal interactive style more oriented toward social and enjoyable aspects, often at the expense of didactic and cognitive elements, typically related to a maternal style [54,55,56]. In line with this, the recent literature also underscored how fathers and mothers tend to structure their children’s activities differently. Specifically, mothers are inclined to develop more vertical interaction aimed at guiding, teaching, and engaging in conversation [57], while fathers are more likely to engage in play activities with their children, acting more as peers and expanding their interaction more horizontally [58]. Horizontal interactions tend to be more reciprocal, cooperative, explorative, and based on shared goals. This interactive role played by fathers promotes children’s intentionality through active encouragement toward initiative taking while maintaining a parental role in guiding the exchange [57,59]. This research evidence seems in line with the findings that emerged from this study. Specifically, changes in fathers’ non-intrusiveness and greater child involvement with fathers may be related to this characteristic paternal style in engaging in horizontal interplays that may lead to more spontaneous and playful interactions. This may also help fathers acquire such skills while adapting to their autistic children. Notably, fathers also showed good abilities in learning to structure their interactions, indicating that they can also improve in developing vertical interactions. Although fathers exhibited greater improvements, mothers also significantly increased their structuring and non-intrusiveness skills during intervention. Similarly, children’s involvement with their mothers also significantly increased during intervention.

Another interesting aspect to consider is related to caregivers’ individual traits that may impact parents’ trajectories of change. Several studies highlighted that mothers and fathers of autistic children usually experience different stress and mental health levels. In particular, mothers seem to experience higher levels of stress [60,61,62,63,64], anxiety, and depression [65,66,67] compared to fathers. Interestingly, mothers’ stress seems to be mitigated by the fathers’ presence in the family system, underlying their fundamental interactional role [68,69]. Given that high levels of stress can significantly compromise the parent–child relationship [70,71], it could be helpful to consider intervention strategies to support both mothers and fathers based on their specific characteristics, strengthening protective factors and enhancing the whole family system.

Interestingly, despite highlighting some differences between the two parents, children showed similar improvement in their abilities to respond during the intervention to their parents, which is in line with the previous literature [43,44]. Responsiveness abilities of children are initially low, and after intervention, they are located in the moderate–low range with both parents after a significant increase.

Child involvement also significantly improved with both parents, although our data indicated a general tendency for children to be more involved when interacting with their fathers.

To conclude, our results highlight changes during the intervention, pointing out both similarities and differences in trajectories of change between parents. These preliminary findings underscore the importance of systematically involving fathers in therapeutic settings, offering the child diverse interactive opportunities to promote the generalization of more advanced and rich interactive skills.

Despite the relevant aspects highlighted in this study, there are several limitations that we need to consider and address in further research. First, the small sample size limits the generalization of our results and the power of these analyses in detecting small to moderate effects. Despite robust estimation techniques, non-significant results should be cautiously interpretated. Because of gender imbalance in our sample, we could not investigate gender differences, and our results may not extend to the general autistic population. The absence of a comparison group also limited the possibility of comparing our results with those of another group attending the intervention without parental involvement. In addition, no measures of individual parents’ traits (e.g., stress, well-being, personality) were employed and should be included in future studies to investigate moderation and mediation effects. To conclude, the inclusion of parent testimony could provide highly relevant and ecologically valid data and constitutes a noteworthy element for future research.

## 5. Conclusions

To conclude, our results underscore the need to study mothers and fathers in terms of the specific aspects that distinguish them and how these characteristics contribute to the dynamics of the whole family system. From a clinical standpoint, parental intervention strategies and designs should be informed and tailored to these aspects. They should include both parents to maximize their impact, leverage both caregivers’ strengths, and identify specific areas for improvement. Finally, fathers’ involvement may also represent a resource to support mothers’ well-being.

## Figures and Tables

**Figure 1 brainsci-15-00133-f001:**
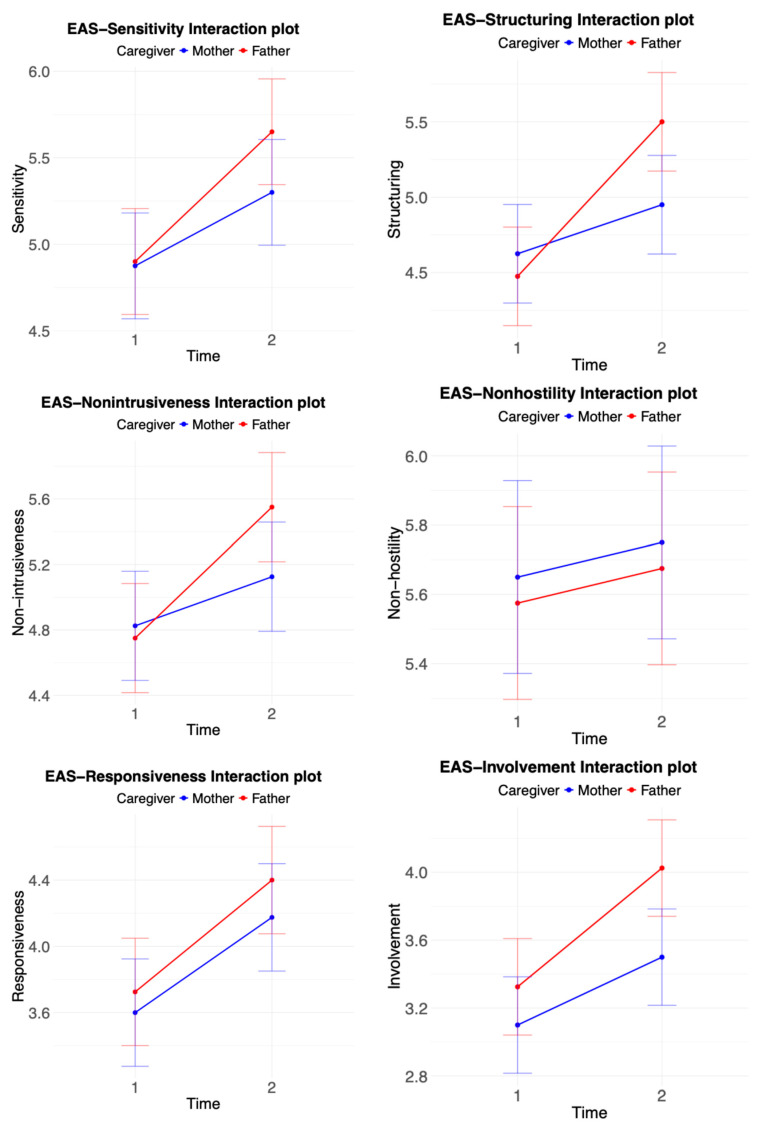
EAS model interaction plots.

**Table 1 brainsci-15-00133-t001:** Descriptive statistics of Emotional Availability Scales.

EA SCALES	Mother Time 1 Mean (DS)	Father Time 1 Mean (DS)	Mother Time 2 Mean (DS)	Father Time 2 Mean (DS)
EA Adult Sensitivity	4.88 (0.86)	4.90 (0.77)	5.30 (0.59)	5.65 (0.43)
EA Adult Structuring	4.63 (0.93)	4.48 (0.77)	4.95 (0.71)	5.50 (0.40)
EA Adult Non-Intrusiveness	4.83 (0.92)	4.75 (0.80)	5.13 (0.74)	5.55 (0.39)
EA Adult Non-Hostility	5.65 (0.65)	5.58 (0.63)	5.75 (0.64)	5.68 (0.52)
EA Child Responsiveness	3.60 (0.80)	3.73 (0.82)	4.18 (0.65)	4.40 (0.53)
EA Child Involvement	3.10 (0.72)	3.33 (0.69)	3.50 (0.54)	4.03 (0.53)

## Data Availability

The data presented in this study are available on request from the corresponding author due to privacy. The data are not publicly available due to privacy reasons.

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
