# Peer review of "Emotional Availability in Autism Intervention: A Mother–Father Comparative Analysis"

_brainsci, 2025, doi:10.3390/brainsci15020133_

Round 1
Reviewer 1 Report
Comments and Suggestions for Authors
Summary: This paper explores the important issue of parental involvement in autism treatment, an area that has been often neglected in past research. With only 14 studies focused on fathers in interventions, the authors analyze changes in interactions involving both parents/caregivers. Using the Emotional Availability Scales (EA), child-mother and child-father play interactions were evaluated at baseline and after 12 months. Results showed significant increases in parental sensitivity, structuring, and non-intrusiveness for both parents, with fathers improving notably in structuring and non-intrusive behaviors. Child responsiveness and involvement also increased. The authors concluded that these interactions became more adaptive, highlighting fathers' receptiveness to developing essential skills. Overall, this paper addresses an important and meaningful question: the logic of the research design and the overall results are reasonable. However, a few places can be further clarified to strengthen this article. I have several questions and suggestions for the authors:
1. In section 1.1, Lines 69 to 72, the authors first introduced the Emotional Availability Scale (ES). I suggest elaborating more on the characteristics and strengths of using EA in this study, as it is the most important measurement in the current study.
2. Contingent on the first comment—In section 2.2.1, the authors explain the Emotional Availability Scale (ES) in long paragraphs. I think many of the descriptions should be moved to the introduction, particularly the history and design of the ES framework. Section 2.2.1 shall only describe the procedural aspects and scoring of the ES.
3. Why did the authors decide to make a one-year intervention? Why not six months or other time periods? The authors shall justify the design even if it is a practical consideration.
4. The authors might want to explain why they tested 40 participants; how was the sample size determined? Was there a power consideration (G*Power calculation)? \
5. I think the information in Table 1 and Figure 1 is redundant, and neither of them follows a standard APA style. For example, all the horizontal grid lines shall be removed except the top and bottom borders. Figure 1 can also be improved—not only is the font size too small, but the data points also lack proper error bars. I suggest careful revisions.
Author Response
Reviewer 1
Comments and Suggestions for Authors
Summary: This paper explores the important issue of parental involvement in autism treatment, an area that has been often neglected in past research. With only 14 studies focused on fathers in interventions, the authors analyze changes in interactions involving both parents/caregivers. Using the Emotional Availability Scales (EA), child-mother and child-father play interactions were evaluated at baseline and after 12 months. Results showed significant increases in parental sensitivity, structuring, and non-intrusiveness for both parents, with fathers improving notably in structuring and non-intrusive behaviors. Child responsiveness and involvement also increased. The authors concluded that these interactions became more adaptive, highlighting fathers' receptiveness to developing essential skills. Overall, this paper addresses an important and meaningful question: the logic of the research design and the overall results are reasonable. However, a few places can be further clarified to strengthen this article. I have several questions and suggestions for the authors:
<Thank you very much for your comments and suggestions on our work. All the comments have been carefully considered, and each will be addressed individually in this letter. We believe that the manuscript has significantly benefited from the feedback received.>>
- In section 1.1, Lines 69 to 72, the authors first introduced the Emotional Availability Scale (ES). I suggest elaborating more on the characteristics and strengths of using EA in this study, as it is the most important measurement in the current study.
- Contingent on the first comment—In section 2.2.1, the authors explain the Emotional Availability Scale (ES) in long paragraphs. I think many of the descriptions should be moved to the introduction, particularly the history and design of the ES framework. Section 2.2.1 shall only describe the procedural aspects and scoring of the ES.
<Thanks for this comment. As you suggested, we moved part of the section 2.2.1 into the introduction to provide more detailed information about the EA construct and theoretical framework. In section 2.2.1 the description is now focused on descriptive elements and procedural aspects and scoring. See all the edits in track changes in the revised version of our manuscript >>
- Why did the authors decide to make a one-year intervention? Why not six months or other time periods? The authors shall justify the design even if it is a practical consideration.
<Thank you for the opportunity to clarify this aspect. The choice of a one-year interval was made for several reasons. On one hand, our laboratory routinely conducts follow-ups and re-administers the full set of tests after one year. The intelligence tests and the ADOS, used for diagnostic purposes, should be administered no more frequently than once yearly. While other tools, such as developmental scales, can be administered as frequently as every three months, we opted for a one-year interval to ensure that all measures in this sample were collected at comparable time points. Moreover, the primary aim of the study was not to track the children's developmental outcomes over time but rather to observe dyadic changes in aspects of affective quality. We believe that a 12-month period provides a reasonable timeframe to capture subtle changes in this domain, considering the structural framework and design of the intervention described in the methods section. We have clarified these considerations in the procedures section to enhance the clarity of our rationale. Thank you again for your thoughtful feedback. >>
- The authors might want to explain why they tested 40 participants; how was the sample size determined? Was there a power consideration (G*Power calculation)?
<2, we provided paired Cohen’s d effect sizes with 95% CIs for the main effects of caregiver and time to help in placing the results in a better perspective.
- I think the information in Table 1 and Figure 1 is redundant, and neither of them follows a standard APA style. For example, all the horizontal grid lines shall be removed except the top and bottom borders. Figure 1 can also be improved—not only is the font size too small, but the data points also lack proper error bars. I suggest careful revisions.
Thank you for your comment. We are aware that both Table 1 and Figure 1 do not follow APA style, but we followed the format of the template provided by the journal. We also added error bars with models’ standard errors and increased the font size. We hope that the figures are now more appropriate. Finally, we included both the Table for an agile reading of EAS dimensions and the Figure plots that include information about variable interaction. >>
Reviewer 2 Report
Comments and Suggestions for Authors
The authors describe a heartening multicomponent parent training treatment for autistic children in Italy. The focus on fathers helps to fill a longtime dearth of knowledge in this area. Readers will be interested to read a follow-up study focusing on which components of this treatment appear to be most impactful -- those that increase children's capacities or those that help parents be more mindful and responsive to their children -- but for the moment, we have a promising treatment that increases child responsiveness and parental emotional availability.
Because this is a small sample, an additional power analysis section and the provision of effect sizes in the results will help readers to place the results into better perspective. Also, because of already limited statistical power, I am unclear why the ANOVAs were done when the LLM's sufficiently address interaction effects.
Parent testimony might have helped bring the treatment more "to life", but I understand if this was not a component of the research. Perhaps this could be cited as another future direction?
Author Response
Reviewer 2
Comments and Suggestions for Authors
The authors describe a heartening multicomponent parent training treatment for autistic children in Italy. The focus on fathers helps to fill a longtime dearth of knowledge in this area. Readers will be interested to read a follow-up study focusing on which components of this treatment appear to be most impactful -- those that increase children's capacities or those that help parents be more mindful and responsive to their children -- but for the moment, we have a promising treatment that increases child responsiveness and parental emotional availability.
Because this is a small sample, an additional power analysis section and the provision of effect sizes in the results will help readers to place the results into better perspective. Also, because of already limited statistical power, I am unclear why the ANOVAs were done when the LLM's sufficiently address interaction effects.
<2 provided for mixed models, we included paired Cohen’s d with 95% CIs as a direct measure of effect size for the main effects of time and caregiver, taking into account within-subject correlation. >>
Parent testimony might have helped bring the treatment more "to life", but I understand if this was not a component of the research. Perhaps this could be cited as another future direction?
<<Dear Reviewer,
Thank you for your comment. We agree that collecting parent testimony could provide highly relevant and ecologically valid data, particularly in the context of interventions involving parental engagement. Following your suggestion, we have included this as a noteworthy element in the directions for future research >>
Round 2
Reviewer 1 Report
Comments and Suggestions for Authors
I thank the authors for carefully addressing my questions and comments. The revised version significantly improved the readability and quality of data presentation. However, I am still not fully satisfied with the current Figure 1. I appreciate the authors adding the error bars; however, the image's resolution is low, the subtitle of each graph is hard to read, and the font size on the X- and Y-axis is too small. I hope the authors can replot a better figure that looks clear and professional.
Author Response
Dear Reviewer,
We provided a revised version of the manuscript with improved image resolution and increased font size to support readability. We hope that the figure is now suitable. Thanks